# Mortality Risk of Sarcopenia and Malnutrition in Older Patients with Type 2 Diabetes Mellitus

**DOI:** 10.3390/nu17162622

**Published:** 2025-08-13

**Authors:** Shinta Yamamoto, Yoshitaka Hashimoto, Fuyuko Takahashi, Ryosuke Sakai, Yuto Saijo, Chihiro Munekawa, Hanako Nakajima, Noriyuki Kitagawa, Rieko Nakatani, Takafumi Osaka, Hiroshi Okada, Naoko Nakanishi, Saori Majima, Emi Ushigome, Masahide Hamaguchi, Michiaki Fukui

**Affiliations:** 1Department of Endocrinology and Metabolism, Graduate School of Medical Science, Kyoto Prefectural University of Medicine, Kyoto 602-0841, Japan; todayweb@koto.kpu-m.ac.jp (S.Y.);; 2Department of Diabetes and Endocrinology, Matsushita Memorial Hospital, Osaka 570-8540, Japan; 3Department of Diabetes and Metabolism, Osaka Railway Hospital, Osaka 545-0053, Japan; 4Department of Diabetology, Kameoka Municipal Hospital, Kameoka 621-0826, Japan; 5Department of Endocrinology and Diabetology, Ayabe City Hospital, Ayabe 623-0011, Japan

**Keywords:** body composition, Cox regression, Geriatric Nutritional Risk Index, handgrip strength, mortality, nutritional risk, older adults, sarcopenia, skeletal muscle mass, type 2 diabetes mellitus

## Abstract

**Aim**: This study aimed to investigate how sarcopenia and nutritional risk influence all-cause mortality among older individuals with type 2 diabetes mellitus. **Methods**: In view of the presence of sarcopenia, defined according to the Asian Working Group for Sarcopenia (AWGS) criteria, and nutritional risk, as determined by the Geriatric Nutritional Risk Index (GNRI), a total of 396 participants were divided into four distinct groups (group 1: no nutritional risk and no sarcopenia, *n* = 306; group 2: nutritional risk and no sarcopenia, *n* = 32; group 3: no nutritional risk and sarcopenia, *n* = 36; and group 4: nutritional risk and sarcopenia, *n* = 22). Mortality risk was assessed through time-to-event analysis using Cox regression. **Results**: Throughout the 86-month median follow-up, 31 participants died. Compared to group 1, hazard ratios (HRs) for mortality of groups 2, 3, and 4 were 9.08 (95% confidence interval (95% CI), 2.44–33.8), 9.08 (95% CI: 2.44–33.8), and 14.0 (95% CI: 4.62–42.4), respectively. The risk of death was significantly higher in groups 2, 3, and 4 compared to group 1. Additionally, group 4 had a significantly higher risk of death than group 3. However, no significant difference in mortality risk was observed between groups 3 and 4 when compared to group 2. **Conclusions**: Coexistence of nutritional risk and sarcopenia was linked to an increased risk of mortality across older individuals with type 2 diabetes mellitus. There was no significant difference in mortality between individuals presenting or not presenting with sarcopenia within the nutritional risk group; therefore, greater attention should be directed toward malnutrition.

## 1. Introduction

Type 2 diabetes mellitus has become a significant global public health issue, particularly in aging populations [1]. In Japan, the prevalence of type 2 diabetes mellitus among older adults continues to rise, reflecting broader demographic shifts and changes in lifestyle patterns [2]. It has been documented that patients diagnosed with type 2 diabetes mellitus exhibit a mortality rate that exceeds that of the general population by 15% [3], highlighting the significance of identifying factors associated with elevated mortality risk in this population.

Sarcopenia, defined as a progressive and generalized decline in skeletal muscle mass and strength, represents a key factor in aging populations. Its prevalence is significantly elevated in individuals with type 2 diabetes mellitus, likely attributable to underlying mechanisms such as insulin resistance, persistent low-grade inflammation, and impaired protein metabolism [4,5]. Our previous research showed that sarcopenia was linked to a six-fold increased mortality risk among individuals aged 60 and over with type 2 diabetes mellitus [6]. Furthermore, sarcopenia and malnutrition are highly interrelated in older individuals [7], and malnutrition has been shown to contribute to an increased incidence of sarcopenia in the general population [8,9,10]. Additionally, both our research and that of others have demonstrated a relationship between sarcopenia and malnutrition in patients with type 2 diabetes mellitus [11,12,13,14].

The Geriatric Nutritional Risk Index (GNRI) is one of several nutritional assessment tools used to evaluate nutrition-related morbidity and mortality. Assessment is based on serum albumin concentrations and the ratio of actual to ideal body weight [15]. The GNRI has been widely used to assess nutritional status in older individuals and has been reported to correlate with hospital readmission, mortality [16], and renal outcomes [17] in various populations. Moreover, changes in GNRI scores over time have demonstrated significant associations with both all-cause and cardiovascular disease (CVD) mortality among patients undergoing dialysis [18].

In patients with type 2 diabetes mellitus, the GNRI has been reported to correlate with adverse renal outcomes [19] and the development of osteosarcopenia [20]. Although some studies have suggested an association between the GNRI and mortality [21], research on this topic, particularly among patients with type 2 diabetes mellitus, remains limited. A previous study examined the impact of malnutrition and sarcopenia on mortality in hospitalized patients with type 2 diabetes mellitus, with a one-year follow-up period [22]; however, no long-term studies have been conducted in outpatient settings. In this study, we extended the follow-up period of the cohort data used in our previous research [6] to investigate how sarcopenia and malnutrition influence mortality risk in patients with type 2 diabetes mellitus.

## 2. Methods

### 2.1. Study Design and Participants

Our previous publication contains the details of the study design [6]. This investigation was carried out as a sub-analysis of the KAMOGAWA-DM cohort study, an ongoing longitudinal study launched in 2014 [23]. From 20 January 2015, to 5 January 2021, patients with type 2 diabetes mellitus were enrolled from the Endocrinology, Diabetes, and Metabolism outpatient clinics at Kyoto Prefectural University of Medicine and Kameoka Municipal Hospital.

Approval for this study was granted by the Ethics Committee of the Kyoto Prefectural University of Medicine (approval number: RBMR-E-466-6), and this study was conducted in accordance with the ethical standards of the 1964 Declaration of Helsinki and its later amendments. Each participant gave written informed consent prior to participation in the study. Type 2 diabetes mellitus was diagnosed by the attending physicians at each clinic based on established diagnostic criteria [24]. Subjects were excluded from the study if they fulfilled any of the predetermined exclusion criteria: (1) missing data on body composition; (2) missing data on handgrip strength; (3) missing covariate data, including pharmacotherapy; lifestyle factors such as physical activity, alcohol consumption, and smoking history; medical history of CVD and malignancy, and metabolic parameters such as HbA1c, triglycerides, and albumin; and (4) age under 60 years [25].

### 2.2. Data Collection

In this study involving human participants, sex was defined as a binary biological variable (male/female) and incorporated as a covariate in all regression models. Gender was not assessed, which may limit interpretation with respect to gender-based factors. The earliest of the following was used to determine the duration of diabetes: self-reported diagnosis, the date of the initiation of diabetes-related treatment, or the date of the first abnormal test result related to diabetes. Lifestyle factors—namely physical activity, tobacco use, and alcohol consumption—were systematically assessed. “Physical activity” was defined as engaging in any form of exercise at least once per week, regardless of the type, frequency, intensity, or duration of the activity. Due to the retrospective nature of the study, detailed information regarding the type and amount of exercise was not available. “Smoking history” was recorded as a binary variable (current smoker or non-smoker). Quantitative data such as pack-years were not collected. “Alcohol consumption” was defined as daily alcohol intake. Data on the history of CVD, including stroke (ischemic or hemorrhagic), heart failure, angina pectoris, coronary artery disease, and prior acute myocardial infarction, as well as malignancy, were retrieved from electronic medical records documented in the hospital information system. Data on the use of medications for diabetes, hypertension, and corticosteroids were also extracted from electronic medical records. Key laboratory parameters, including HbA1c, triglycerides, and serum albumin, were measured from venous blood samples drawn following an overnight fast. Handgrip strength was measured in both hands using a handgrip dynamometer (Smedley type, TKK, Takei Scientific Instruments, Niigata, Japan), and the maximum recorded value was used for analysis [26]. Body weight and appendicular skeletal muscle mass were assessed using a multifrequency bioelectrical impedance analyzer (InBody 720; InBody Japan, Tokyo, Japan). Subsequently, body mass index (BMI) and skeletal muscle mass index (SMI) were calculated based on the following formulas.

BMI (kg/m^2^) = weight (kg) ÷ height^2^ (m^2^),SMI (kg/m^2^) = appendicular muscle mass (kg) ÷ height^2^ (m^2^).

### 2.3. Definitions of Sarcopenia

Sarcopenia was diagnosed based on the guidelines of the Asian Working Group for Sarcopenia [27]. The cutoff values for low handgrip strength were established as <28 kg for men and <18 kg for women, while the cutoff values for low SMI were established as <7.0 kg/m^2^ for men and <5.7 kg/m^2^ for women. Participants who satisfied both aforementioned criteria were diagnosed with sarcopenia. Although gait speed is also a diagnostic criterion for sarcopenia, it was not included in this study due to the lack of relevant measurements.

### 2.4. Definitions of GNRI

The GNRI was determined using the following formula:GNRI = (14.89 × serum albumin) + (41.7 × [body weight/ideal body weight]) [15].

Ideal body weight was derived using the Lorentz formula [28]: ideal weight = height (cm) − 100 − [(height − 150)/4] (men) or height (cm) − 100 − [(height − 150)/2.5] (women). Participants were classified into two groups based on the GNRI: “nutritional risk” (GNRI ≤ 98) and “no nutritional risk” (GNRI > 98) [15].

### 2.5. Outcome of This Study and Follow-Up

The primary endpoint of this study was all-cause mortality, ascertained via electronic medical records. Follow-up time, measured in months, spanned from the baseline assessment to the occurrence of death, last recorded contact, or relocation to another healthcare institution. Causes of death were categorized as follows:CVD-related mortality (stroke [ischemic or hemorrhagic], heart failure, angina pectoris, coronary artery disease, or past acute myocardial infarction),Malignancy-related mortality (solid or hematologic malignancies),Pneumonia-related mortality (including aspiration pneumonia),Other causes.

### 2.6. Statistical Analysis

Statistical analyses were executed using R software (version 4.4.1; R Foundation for Statistical Computing, Vienna, Austria) alongside RStudio (2024.04.1+748). Statistical significance was defined as a *p*-value less than 0.05. Since this study was conducted as a retrospective analysis of data obtained from the KAMOGAWA-DM cohort, a formal a priori sample size calculation was not performed. Instead, all patients with available and complete data who met the inclusion criteria between January 2015 and January 2021 were included in the analysis. This approach was chosen to ensure comprehensive inclusion of eligible participants, thereby maximizing statistical power and minimizing selection bias within the constraints of the available dataset. To assess the adequacy of the sample size, a post hoc power analysis was conducted. Given the total sample size of 396 and 37 observed deaths, the study had 97.6% power to detect a hazard ratio of 2.5 at a two-sided alpha level of 0.05. This effect size was smaller than the hazard ratios observed in our primary analysis, supporting the robustness of the study’s findings. Patients were categorized into four groups in accordance with the presence or absence of nutritional risk and sarcopenia: **group 1:** no nutritional risk and no sarcopenia, **group 2:** nutritional risk and no sarcopenia, **group 3:** no nutritional risk and sarcopenia, and **group 4:** nutritional risk and sarcopenia. Categorical variables were expressed as frequencies and percentages. For continuous variables, data were reported as mean ± standard deviation (SD) when normally distributed. Hazard ratios (HRs) and 95% confidence intervals (95% CIs) for all-cause mortality were estimated using Cox proportional hazards models. A Cox proportional hazards model was constructed to compare the four groups, and pairwise comparisons between each group were conducted, resulting in six Cox proportional hazards models.

Three models were developed for the analyses: model 1: unadjusted (no covariates included), model 2: adjusted for age and sex, and model 3: further adjusted for additional variables, including duration of diabetes, HbA1c, triglyceride levels, history of CVD, history of malignancy, smoking history, alcohol consumption, and exercise habits. The covariates included in model 3 were selected based on both theoretical considerations and prior empirical evidence: age and sex (universal risk factors for mortality), duration of diabetes and HbA1c (markers of diabetes severity), triglyceride levels (related to nutritional status and cardiovascular risk) [29], history of CVD and malignancy (strong predictors of mortality), smoking and alcohol consumption (lifestyle factors with established health impacts), exercise habits (protective lifestyle factor) [29,30,31], and the use of SGLT2 inhibitors, GLP-1 receptor agonists, insulin, and corticosteroids (medications that may affect metabolic status, muscle mass, body composition, and clinical outcomes) [32,33,34]. A *p* value ≥ 0.05 was interpreted as evidence that the proportional hazards assumption was met. As a sensitivity analysis, we assessed the effect of nutritional risk and sarcopenia on mortality after excluding patients with malignancy status, using the same model framework described above. Survival analysis was conducted using the Kaplan–Meier method to illustrate differences in mortality according to nutritional risk and sarcopenia status. The log-rank test was employed to assess statistical significance between the resulting survival curves.

## 3. Results

Between 20 January 2015 and 5 January 2021, data from 702 patients with type 2 diabetes mellitus were extracted from the database. Following the exclusion of 306 individuals based on predefined criteria, 396 participants were included in the final analysis (Figure 1). Among them, 232 (58%) were male. The mean age was 71.3 years (SD: 6.3), with a mean diabetes duration of 16.3 years (SD: 11.3) and a mean HbA1c level of 56 mmol/mol (SD: 10.9).

Participants were classified into four groups according to their nutritional risk (GNRI ≤ 98 indicating risk, >98 indicating no risk) and sarcopenia status. The group distribution was as follows (Table 1):Group 1: no nutritional risk, no sarcopenia (*n* = 306),Group 2: nutritional risk, no sarcopenia (*n* = 32),Group 3: no nutritional risk, with sarcopenia (*n* = 36),Group 4: both nutritional risk and sarcopenia (*n* = 22).

**Table 1 nutrients-17-02622-t001:** Clinical characteristics of study participants.

	ALL (N = 396)	Group 1 (N = 306)	Group 2 (N = 32)	Group 3 (N = 36)	Group 4 (N = 22)
Sex (men/women)	232/164	175/131	20/12	23/13	14/8
Age (years)	71.3 (6.3)	70.2 (5.8)	71.2 (5.5)	76.5 (5.7)	78.6 (5.4)
GNRI	106.0 (11.7)	110.0 (6.6)	86.9 (17.2)	104.0 (5.1)	86.4 (17.9)
Duration of diabetes (years)	16.3 (11.3)	15.3 (10.9)	16.7 (12.0)	23.6 (12.6)	17.8 (9.7)
Family history of diabetes (−/+)	238/158	181/125	17/15	21/15	19/3
Height (cm)	160.6 (8.7)	161.0 (8.8)	161.8 (8.0)	157.3 (9.0)	158.2 (7.1)
Body weight (kg)	61.1 (10.8)	63.6 (10.3)	53.4 (6.5)	53.8 (8.6)	49.5 (7.4)
Handgrip strength (kg)	27.1 (8.7)	28.5 (8.6)	26.8 (7.8)	20.3 (6.4)	18.9 (5.6)
Skeletal muscle index (kg/m^2^)	6.9 (1.0)	7.1 (0.9)	6.6 (0.9)	6.0 (0.8)	5.9 (0.8)
Body mass index (kg/m^2^)	23.6 (3.7)	23.6 (3.7)	20.5 (2.8)	21.7 (2.7)	19.7 (1.9)
Presence of hypertension (−/+)	121/275	94/212	10/12	10/26	7/15
SGLT2 inhibitor (−/+)	339/57	255/51	31/1	33/3	20/2
GLP-1 receptor agonist (−/+)	342/54	266/40	26/6	30/6	20/2
Insulin (−/+)	295/101	236/70	18/14	25/11	16/6
Corticosteroids (−/+)	382/14	301/5	29/3	34/2	18/4
History of heart disease (−/+)	320/76	258/48	27/5	23/13	12/10
History of malignancy (−/+)	331/65	257/49	26/6	29/7	19/3
Habit of smoking (−/+)	337/59	260/46	26/6	33/3	18/4
Habit of exercise (−/+)	198/198	158/148	17/15	13/23	10/12
Habit of drinking alcohol (−/+)	277/119	216/90	20/12	22/14	19/3
Hemoglobin A1c (mmol/mol)	56 (10.9)	56 (10.9)	60 (19.7)	57 (9.8)	58 (13.1)
Plasma glucose (mmol/L)	8.2 (2.6)	8.1 (2.5)	8.9 (4.0)	8.2 (1.9)	9.1 (3.2)
Triglycerides (mmol/L)	1.4 (0.9)	1.5 (1.0)	1.2 (0.7)	1.3 (0.6)	1.3 (0.6)
Albumin (g/dL)	4.2 (0.3)	4.3 (0.3)	3.7 (0.4)	4.3 (0.2)	3.7 (0.4)

Data were expressed as the mean (standard deviation) or a number. Participants were categorized into four groups based on their nutritional risk and the presence of sarcopenia. Group 1 included individuals with no nutritional risk and no sarcopenia. Group 2 consisted of individuals with nutritional risk but without sarcopenia. Group 3 included those with no nutritional risk but with sarcopenia. Group 4 consisted of individuals with both nutritional risk and sarcopenia. The following abbreviations are used: SGLT2, sodium-glucose co-transporter 2; GLP-1, glucagon-like peptide-1; GNRI, Geriatric Nutritional Risk Index.

By the end of the follow-up period (31 December 2024), 37 participants (9%) had died. Malignancy was identified as the leading cause of death, accounting for 15 cases (41%) (Table 2). The median follow-up duration was 86 months. Survival analysis using Kaplan–Meier curves (Figure 2) and Cox proportional hazards models indicated a significant association between mortality and the presence of both nutritional risk and sarcopenia (log-rank test, *p* < 0.001).

After adjustment for relevant covariates, participants in group 4 exhibited a significantly greater risk of death compared to those in group 1 (HR: 13.1, 95% CI: 4.97–34.6, *p* < 0.001). Similarly, groups 2 and 3 also had elevated mortality risks relative to group 1 (group 2: HR 7.71, 95% CI: 2.60–22.9, *p* < 0.001; group 3: HR 3.86, 95% CI: 1.29–11.6, *p* = 0.02) (Table 3).

To compare survival across all groups, six pairwise Cox regression models were performed, and corresponding Kaplan–Meier curves were generated (Figure 3). The log-rank tests revealed statistically significant mortality differences among all groups, except between groups 2 and 3 (*p* < 0.05). In multivariable-adjusted models, groups 2 (HR: 9.08, 95% CI: 2.44–33.8, *p* = 0.001), 3 (HR: 9.08, 95% CI: 2.44–33.8, *p* = 0.001), and 4 (HR: 14.0, 95% CI: 4.62–42.4, *p* < 0.001) had significantly higher mortality risks than group 1. Furthermore, mortality was significantly higher in group 4 than in group 3 (HR: 3.69, 95% CI: 1.15–11.9, *p* = 0.03), while no significant difference was observed between groups 4 and 2 (HR: 0.93, 95% CI: 0.16–5.33, *p* = 0.94) (Table 4).

A sensitivity analysis excluding participants with malignancy produced consistent results. Kaplan–Meier analysis and Cox regression (Figure 4) confirmed that both sarcopenia and nutritional risk were significantly associated with higher mortality risk (log-rank test, *p* < 0.001). Among participants without malignancy, groups 2, 3, and 4 showed significantly increased mortality risks compared to group 1, with hazard ratios of 5.32 (95% CI: 1.07–26.5, *p* = 0.04), 12.12 (95% CI: 3.27–44.9, *p* < 0.001), and 27.11 (95% CI: 6.39–115.0, *p* < 0.001), respectively (Table 5).

## 4. Discussion

This present study demonstrated that both nutritional risk, as measured by the GNRI, and the presence of sarcopenia were associated with an elevated risk of mortality in older patients with type 2 diabetes mellitus. After adjusting for covariates, both nutritional risk and sarcopenia were independently associated with an increased risk of mortality (graphical abstracts). This finding was supported by the significant differences observed between group 1 and groups 2, 3, and 4. Moreover, mortality risk was higher in individuals with nutritional risk regardless of sarcopenia status (group 1 vs. group 2 and group 3 vs. group 4). Conversely, among those without nutritional risk, the presence of sarcopenia was associated with increased mortality (group 1 vs. group 3), whereas, among those with nutritional risk, there was no significant difference in mortality risk between those with and without sarcopenia (group 2 vs. group 4). These results remained significant even after adjusting for malignancy as a covariate, and similar findings were observed when participants with malignancy were excluded. Notably, similar associations between nutritional risk, sarcopenia, and mortality were also observed in model 2, which adjusted only for age and sex. These results, which are consistent across various models and sensitivity analyses, support the robustness of our findings.

Previous studies have reported similar associations. For instance, our previous study found that the presence of sarcopenia increased mortality risk by 6.12-fold (95% CI: 1.52–24.7) [6]. Another study reported a 2.58-fold (95% CI: 1.67–3.99) increase in mortality risk associated with nutritional risk as assessed by the GNRI [21]. Moreover, earlier studies had shown that the combination of sarcopenia and nutritional risk was associated with a 2.01-fold increase in one-year mortality risk among in hospitalized patients with type 2 diabetes mellitus. Consistent with these findings, our study, which featured a longer follow-up period with a median duration of 86 months, demonstrated that the combined presence of sarcopenia and nutritional risk was associated with a 13.1-fold increase in mortality risk. This finding underscores the significance of addressing both sarcopenia and malnutrition in long-term care for older individuals with type 2 diabetes mellitus.

Patients with type 2 diabetes mellitus are more prone to developing sarcopenia [4,35]. Sarcopenia has been demonstrated to be associated with elevated oxidative stress, reduced antioxidant capacity [36], and inflammatory responses [37], which collectively contribute to decreased protein synthesis and increased protein degradation, ultimately leading to an elevated risk of mortality [38]. The GNRI provides a practical approach to evaluating nutritional status using serum albumin and BMI. It has been reported to demonstrate higher sensitivity and specificity compared to other nutritional screening tools in evaluating nutritional status in maintenance dialysis patients [39]. Additionally, the GNRI has been shown to more accurately predict CVD mortality compared to serum albumin or BMI alone in patients with dialysis [40], highlighting its value as a robust marker for nutritional assessment. Nutritional status has been widely recognized as a prognostic factor in various diseases [17,41,42,43], and in patients with chronic conditions such as diabetes, the risk of malnutrition is known to be elevated [44,45]. As noted above, sarcopenia and malnutrition frequently coexist in patients with type 2 diabetes mellitus. These conditions not only are individually associated with mortality risk but also act additively, with the coexistence of both conditions significantly amplifying the risk of mortality. This finding is particularly important and underscores the need for comprehensive assessments of both sarcopenia and nutritional status in older individuals with type 2 diabetes mellitus.

In this study, we found that while there was a statistically significant difference in the mortality risk between patients with and without nutritional risk, there was no statistically significant difference between the mortality risk of patients with sarcopenia and the mortality risk of patients without sarcopenia among patients with nutritional risk. These findings suggest that nutritional risk alone substantially contributes to mortality risk, potentially diminishing the relative influence of sarcopenia under these conditions.

One possible explanation is that malnutrition affects mortality through mechanisms other than those associated with muscle loss due to sarcopenia, such as metabolic dysfunction and reduced anti-inflammatory capacity. Malnutrition itself is a known contributor to the development of sarcopenia [46], and participants in the group with nutritional risk but without sarcopenia were likely at higher risk of developing sarcopenia over time. Additionally, malnutrition-induced impairments in lymphocyte function and reductions in natural host defense mechanisms, such as macrophages and granulocytes [47], may further contribute to the increased mortality risk observed in patients with malnutrition. These mechanisms, while plausible, warrant further investigation to elucidate the precise pathways underlying these associations.

The strengths of this study can be summarized as follows. First, while previous studies evaluating the impact of sarcopenia and malnutrition on mortality risk in patients with diabetes were limited to inpatients and a relatively short-term follow-up period of approximately one year, our study adopted outpatients and a long-term follow-up period with a median duration of 86 months (approximately seven years), allowing for a more comprehensive analysis. Second, both sarcopenia and nutritional risk as assessed by the GNRI can be evaluated using simple and accessible methods, and our findings highlight their potential utility in predicting mortality risk, thereby demonstrating high clinical applicability. However, this study has several limitations. First, the present study’s population exclusively comprised Japanese individuals, a factor that may serve to limit the generalizability of the findings to other ethnic groups. Second, the relatively small number of mortality events necessitates further follow-up studies to validate our results. Third, there was a notable imbalance in group sizes (for example, group 1: N = 306 vs. group 4: N = 22). The small sample sizes in some groups may have resulted in limited statistical power and unstable estimates, leading to wider confidence intervals. Finally, as an observational study, there is a possibility of residual confounding factors that were not accounted for in the analysis. Additionally, multiple pairwise comparisons were performed in our analysis without adjustment for multiplicity, as the analyses were exploratory and hypothesis-generating in nature. We acknowledge this as a limitation, and future studies should consider appropriate correction for multiple comparisons to minimize the risk of Type I error. In addition, while multiple covariates were included to adjust for potential confounding, future studies with larger sample sizes should consider parsimonious modeling strategies or the use of dimension reduction and penalized regression techniques to minimize the risk of overfitting. It is recommended that subsequent research endeavors seek to address the identified limitations and thereby furnish additional insights into the intricate interplay between sarcopenia, malnutrition, and mortality risk in patients with type 2 diabetes mellitus.

## 5. Conclusions

The coexistence of nutritional risk and sarcopenia was associated with an increased risk of mortality among older individuals with type 2 diabetes mellitus. Furthermore, mortality was elevated in individuals with nutritional risk regardless of whether sarcopenia was present. Conversely, among patients with sarcopenia, the mortality risk exhibited variability depending on their nutritional status, and no significant difference in mortality was observed between those with and without sarcopenia among the individuals with nutritional risk.

The present findings underscore the significance of addressing both sarcopenia and malnutrition in clinical practice; however, particular attention should be paid to malnutrition, as it appears to exert a stronger influence on mortality risk.

## Figures and Tables

**Figure 1 nutrients-17-02622-f001:**
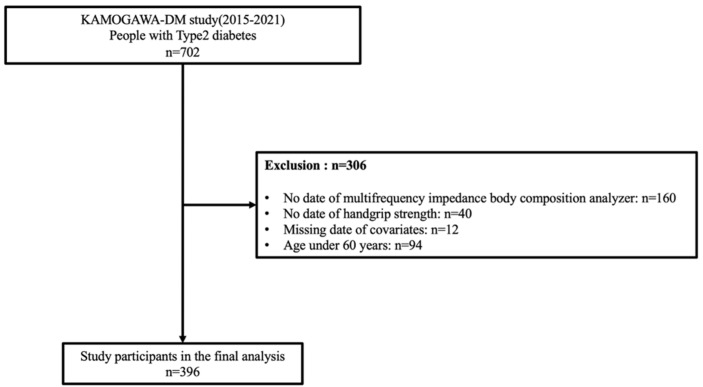
Flow diagram showing the selection of the study population.

**Figure 2 nutrients-17-02622-f002:**
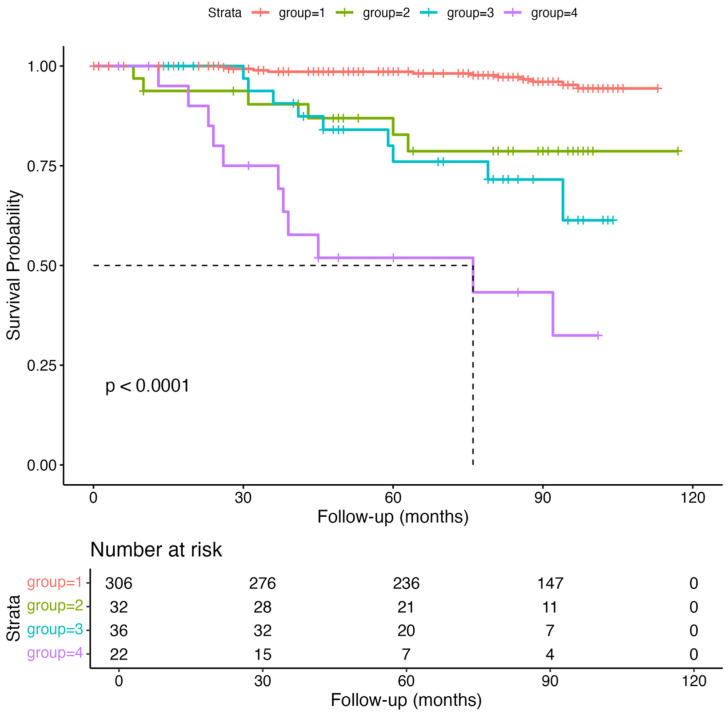
Kaplan–Meier survival curves stratified by nutritional risk and sarcopenia status.

**Figure 3 nutrients-17-02622-f003:**
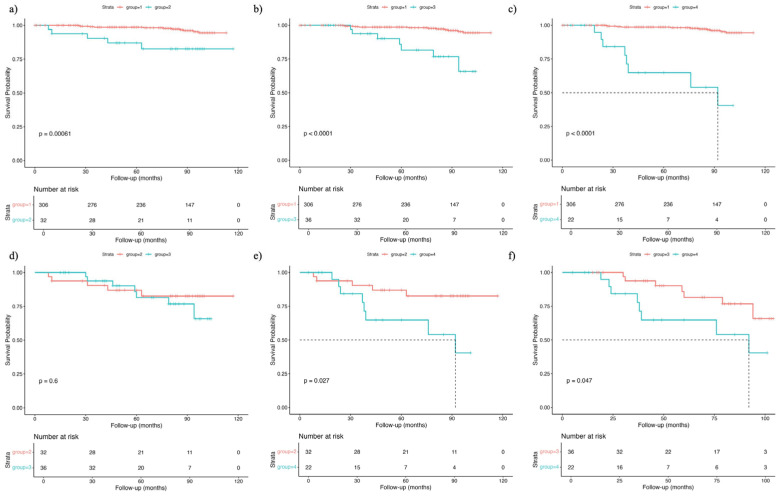
Kaplan–Meier survival curves and log-rank tests were used to compare mortality risk between each pair of the four groups categorized by nutritional risk and sarcopenia status. (**a**) Group 1 vs. group 2, (**b**) group 1 vs. group 3, (**c**) group 1 vs. group 4, (**d**) group 2 vs. group 4, (**e**) group 2 vs. group 3, (**f**) group 3 vs. group 4.

**Figure 4 nutrients-17-02622-f004:**
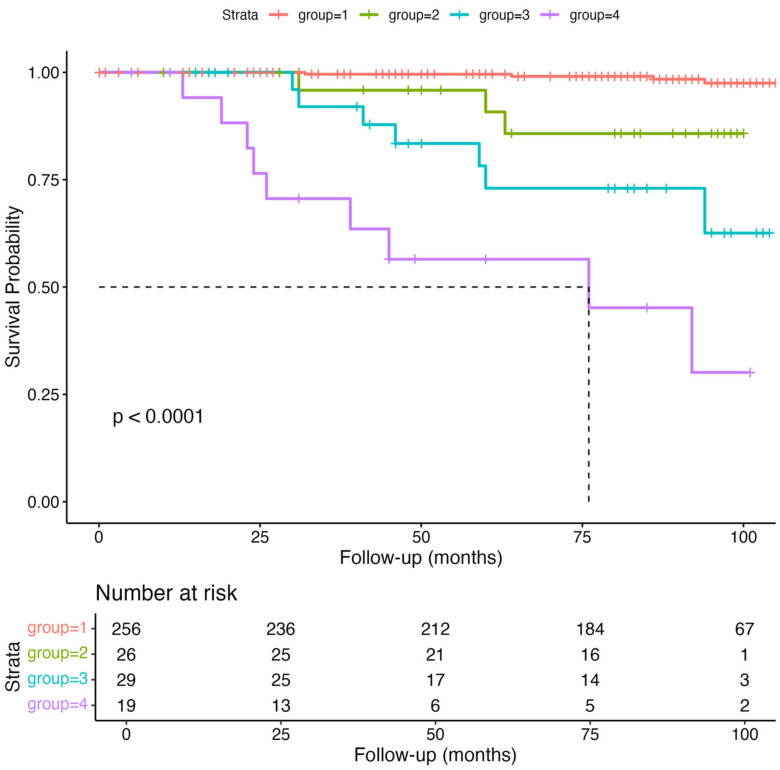
Kaplan–Meier survival curves among participants without a history of malignancy.

**Table 2 nutrients-17-02622-t002:** Details of causes of death.

			Cause of Death		
	Total	Malignancy	CVD	Pneumonia	Other
ALL	37	15	7	3	12
Group 1	12	6	1	0	5
Group 2	6	3	1	0	2
Group 3	9	3	2	2	2
Group 4	11	3	3	1	4

Data were expressed as a number. Malignancy was defined as diseases categorized into solid tumors and hematologic malignancies. Cardiovascular disease (CVD) was defined as conditions including cerebrovascular diseases, such as cerebral hemorrhage and cerebral infarction, as well as heart diseases, such as myocardial infarction and angina pectoris. Pneumonia was defined to include aspiration pneumonia.

**Table 3 nutrients-17-02622-t003:** Cox proportional hazards models stratified by nutritional risk and sarcopenia status.

	Model 1		Model 2		Model 3	
	HR	*p*	HR	*p*	HR	*p*
Group 1 (N = 306)	Ref	-	Ref	-	Ref	-
Group 2 (N = 32)	6.18 (2.28–16.7)	<0.001	5.79 (2.13–15.7)	<0.001	5.75 (2.00–16.5)	0.001
Group 3 (N = 36)	9.14 (3.77–22.2)	<0.001	4.51 (1.70–12.0)	0.002	3.88 (1.39–10.8)	0.009
Group 4 (N = 22)	25.7 (11.0–59.7)	<0.001	9.89 (3.67–26.6)	<0.001	12.0 (4.30–33.3)	<0.001

The Cox proportional hazards models were constructed based on four groups categorized by the presence or absence of nutritional risk and sarcopenia. Group 1 was used as the reference group, and the results are presented as hazard ratios (HRs) (95% confidence intervals). Model 1 represents the crude analysis. Model 2 adjusts for age and sex. Model 3 further adjusts for a history of diabetes, malignancy, cardiovascular diseases, smoking history, alcohol consumption, exercise habits, HbA1c levels, triglyceride levels, and the use of SGLT2 inhibitors, GLP-1 receptor agonists, insulin, and corticosteroids.

**Table 4 nutrients-17-02622-t004:** Pairwise Cox proportional hazards models for groups stratified by nutritional risk and sarcopenia status.

	Model 1		Model 2		Model 3	
	HR	*p*	HR	*p*	HR	*p*
Group 1, group 2	6.32 (2.33–17.2)	<0.001	6.15 (2.25–16.8)	<0.001	5.62 (1.69–18.6)	0.005
Group 1, group 3	9.81 (4.02–23.9)	<0.001	3.88 (1.40–10.7)	0.009	3.39 (1.46–10.9)	0.007
Group 1, group 4	25.5 (10.9–59.7)	<0.001	10.8 (3.68–31.7)	<0.001	14.2 (4.12–49.2)	<0.001
Group 2, group 3	1.46 (0.52–4.10)	0.48	0.72 (0.20–2.56)	0.62	0.63 (0.14–2.76)	0.54
Group 2, group 4	3.83 (1.41–10.4)	0.009	2.44 (0.68–8.76)	0.17	2.44 (0.59–11.9)	0.27
Group 3, group 4	2.87 (1.18–6.95)	0.02	2.56 (1.03–6.33)	0.04	4.39 (1.26–15.3)	0.02

Cox proportional hazards models were developed by comparing two groups at a time from the four groups categorized based on the presence or absence of nutritional risk and sarcopenia. The risks of the group listed on the right side with the group listed on the left side as the control are indicated. The results are presented as hazard ratios (95% confidence intervals). Model 1 represents the crude analysis. Model 2 adjusts for age and sex. Model 3 includes further adjustments for diabetes history, malignancy, cardiovascular diseases, smoking history, alcohol consumption, exercise habits, HbA1c levels, triglyceride levels, and the use of SGLT2 inhibitors, GLP-1 receptor agonists, insulin, and corticosteroids.

**Table 5 nutrients-17-02622-t005:** Cox proportional hazards models for mortality stratified by nutritional risk and sarcopenia status, excluding malignancy status.

	Model 1		Model 2		Model 3	
	HR	*p*	HR	*p*	HR	*p*
Group 1	Ref	–	Ref	–	Ref	-
Group 2	8.26 (1.84–37.0)	0.006	7.51 (1.68–33.6)	0.008	6.29 (1.20–32.9)	0.03
Group 3	20.7 (6.05–71.1)	<0.001	10.7 (2.77–41.2)	<0.001	13.3 (3.42–51.4)	<0.001
Group 4	61.5 (18.7–202.2)	<0.001	19.2 (4.60–79.9)	<0.001	34.0 (7.04–164.5)	<0.001

Cox proportional hazards models were constructed to analyze mortality among the four groups categorized based on the presence or absence of nutritional risk and sarcopenia, excluding patients with malignancy status. The results are presented as hazard ratios (HRs) (95% confidence intervals). Model 1 represents the crude analysis. Model 2 adjusts for age and sex. Model 3 includes further adjustments for diabetes history, cardiovascular diseases, smoking history, alcohol consumption, exercise habits, HbA1c levels, triglyceride levels, and the use of SGLT2 inhibitors, GLP-1 receptor agonists, insulin, and corticosteroids.

## Data Availability

Data from this study are available from the corresponding author upon reasonable request.

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
