# Peer review of "Mortality Risk of Sarcopenia and Malnutrition in Older Patients with Type 2 Diabetes Mellitus"

_nutrients, 2025, doi:10.3390/nu17162622_

Round 1
Reviewer 1 Report
Comments and Suggestions for Authors
The article “Mortality Risk of Sarcopenia and Malnutrition in Older Patients with Type 2 Diabetes Mellitus” is well-written. Here are some concerns that I have regarding the paper.
Although the topic is not new and there is published literature,
My major concern is that the group sizes are highly imbalanced (e.g., Group 1: N=306 vs. Group 4: N=22), which raises concerns about the statistical power and stability of estimates for the smaller groups. Such an imbalance can lead to inflated standard errors, resulting in wide confidence intervals in the paper. Moreover, there is no correction for multiple pairwise comparisons.
Table 1 includes medications such as SGLT2 inhibitors, GLP-1 receptor agonists, corticosteroids, and insulin. It would be helpful if the authors could clarify the rationale for including these specific variables.
Are these medications being examined as potential confounders, markers of disease severity, or therapeutic modifiers of the relationship between nutritional risk/sarcopenia and mortality? Do the authors consider that triglyceride levels may confound the association between the combined nutritional risk/sarcopenia groups and mortality, even after adjusting for history of cardiovascular disease?
Lines 299 and 300 add confidence intervals in the discussion.
Overall, the manuscript can be improved by:
Justify the inclusion of all covariates based on theoretical or empirical considerations, or
consider parsimonious modeling strategies (e.g., adjusting only for strong confounders identified a priori), or
Use dimension reduction or penalized regression methods to mitigate overfitting risks.
Author Response
- My major concern is that the group sizes are highly imbalanced (e.g., Group 1: N=306 vs. Group 4: N=22), which raises concerns about the statistical power and stability of estimates for the smaller groups. Such an imbalance can lead to inflated standard errors, resulting in wide confidence intervals in the paper. Moreover, there is no correction for multiple pairwise comparisons.
Response:
Thank you for your important comment regarding the imbalance in group sizes and the lack of correction for multiple comparisons. We acknowledge that the smaller sample sizes in some groups (e.g., Group 4: N=22) may limit statistical power and result in wider confidence intervals. This limitation has now been addressed and clearly discussed in the revised manuscript. Regarding multiple pairwise comparisons, our analysis was exploratory and hypothesis-generating. However, we recognize this limitation and have now acknowledged it in the discussion.
- Discussion:
However, this study has several limitations. First, the present study's population was exclusively comprised of Japanese individuals, a factor which may serve to limit the generalizability of the findings to other ethnic groups. Second, the relatively small number of mortality events necessitates further follow-up studies to validate our results. Third, there was a notable imbalance in group sizes (for example, Group 1: N=306 vs. Group 4: N=22). The small sample sizes in some groups may have resulted in limited statistical power and unstable estimates, leading to wider confidence intervals. Finally, as an observational study, there is a possibility of residual confounding factors that were not accounted for in the analysis. Additionally, multiple pairwise comparisons were performed in our analysis without adjustment for multiplicity, as the analyses were exploratory and hypothesis-generating in nature. We acknowledge this as a limitation, and future studies should consider appropriate correction for multiple comparisons to minimize the risk of Type I error. It is recommended that subsequent research endeavors seek to address the identified limitations and thereby furnish additional insights into the intricate interplay between sarcopenia, malnutrition, and mortality risk in patients with type 2 diabetes mellitus.
- Table 1 includes medications such as SGLT2 inhibitors, GLP-1 receptor agonists, corticosteroids, and insulin. It would be helpful if the authors could clarify the rationale for including these specific variables. Are these medications being examined as potential confounders, markers of disease severity, or therapeutic modifiers of the relationship between nutritional risk/sarcopenia and mortality? Do the authors consider that triglyceride levels may confound the association between the combined nutritional risk/sarcopenia groups and mortality, even after adjusting for history of cardiovascular disease?
Response:
Thank you for your valuable comment. In Table 1, we presented medications such as SGLT2 inhibitors, GLP-1 receptor agonists, corticosteroids, and insulin to describe the medical background and treatment status of the participants. These medications were not included as covariates in the initial models (Models 1 and 2); however, in response to your comment, we revised Model 3 in the main text to include these medications as covariates. These drugs may reflect disease severity and potentially influence both nutritional status and clinical outcomes, making their adjustment methodologically relevant. The associations between nutritional risk, sarcopenia, and mortality remained largely unchanged after this revision.
Additionally, triglyceride levels were adjusted for only in the revised Model 3. Triglycerides are influenced by nutritional and metabolic factors and are also established markers of cardiovascular risk, regardless of the presence or absence of a history of cardiovascular disease. Therefore, we considered triglyceride levels to be important potential confounders when examining the associations of nutritional risk and sarcopenia with mortality and included them accordingly in Model 3.
The details of the revised Model 3 are presented in the main text.
- Lines 299 and 300 add confidence intervals in the discussion.
Response:
Thank you for your helpful comment.
We have revised the discussion section (lines 299 and 300) to include the relevant confidence intervals, as suggested.
- Discussion:
Previous studies have reported similar associations. For instance, our previous study found that the presence of sarcopenia increased mortality risk by 6.12-fold (95% CI: 1.52–24.7)[6]. Another study reported a 2.58-fold (95% CI: 1.67–3.99) increase in mortality risk associated with nutritional risk as assessed by GNRI[21].
- Justify the inclusion of all covariates based on theoretical or empirical considerations, or consider parsimonious modeling strategies (e.g., adjusting only for strong confounders identified a priori), or Use dimension reduction or penalized regression methods to mitigate overfitting risks.
Response:
Thank you for your helpful suggestions. We have revised the manuscript to better justify the inclusion of each covariate based on theoretical and empirical considerations. We also examined the robustness of our findings by conducting sensitivity analyses, including models with only key covariates such as age and sex, as well as analyses adjusting for or excluding participants with malignancy. These analyses consistently supported our main results. We also discuss the potential advantages of parsimonious modeling and dimension reduction techniques in the limitations section, noting that future studies with larger sample sizes may benefit from these approaches to reduce the risk of overfitting.
- Methods
2.6. Statistical Analysis
Three models were developed for the analyses: model 1: Unadjusted (no covariates included); model 2: Adjusted for age and sex; and model 3: Further adjusted for additional variables, including duration of diabetes, HbA1c, triglyceride levels, history of CVD, history of malignancy, smoking history, alcohol consumption, exercise habits. The covariates included in model 3 were selected based on both theoretical considerations and prior empirical evidence: age and sex (universal risk factors for mortality), duration of diabetes and HbA1c (markers of diabetes severity), triglyceride levels (related to nutritional status and cardiovascular risk), history of CVD and malignancy (strong predictors of mortality), smoking and alcohol consumption (lifestyle factors with established health impacts), and exercise habits (protective lifestyle factor) [29–31].
- Discussion:
Conversely, among those without nutritional risk, the presence of sarcopenia was associated with increased mortality (group 1 vs. group 3), whereas, among those with nutritional risk, there was no significant difference in mortality risk between those with and without sarcopenia (group 2 vs. group 4). These results remained significant even after adjusting for malignancy as a covariate, and similar findings were observed when participants with malignancy were excluded. Notably, similar associations between nutritional risk, sarcopenia, and mortality were also observed in model 2, which adjusted only for age and sex. These consistent results across various models and sensitivity analyses support the robustness of our findings. [...]
In addition, while multiple covariates were included to adjust for potential confounding, future studies with larger sample sizes should consider parsimonious modeling strategies or the use of dimension reduction and penalized regression techniques to minimize the risk of overfitting. It is recommended that subsequent research endeavors seek to address the identified limitations and thereby furnish additional insights into the intricate interplay between sarcopenia, malnutrition, and mortality risk in patients with type 2 diabetes mellitus.
Reviewer 2 Report
Comments and Suggestions for Authors
In this paper, the Authors investigate the effects of both nutritional risk, assessed via a simple score the (GNRI), and sarcopenia on mortality in an elderly population with Type II Diabetes Mellitus, following up to their previous research efforts. Of note, the longer follow up period highlighted the greater impact of nutritional issues on mortality, a problem that directly plays into recent research highlighting dietary patterns and gut microbiota in (healthy) aging.
The study appears well written and adequately circumstantiated by a valid statistical model, although I concur with the Authors that the cohort characteristics and results would benefit from a multicentric, international pool of subjects as well as longer still follow-up times. I have only minor remarks on the text:
Lines 20-21: this sentence, which I assume summarizes inclusion criteria for sarcopenia and nutritional issues, is incorrectly written. Please revise to better clarify, adding a reference to the GNRI which I assume is the metric being discussed.
Lines 118-121: how were physical activity and smoking habit measured? "Any form of exercise on a weekly basis" is a rather generic catch-all, seeing as it can greatly vary from a mere tranquil walk to endurance training. In a similar fashion, were "packets per year" used as a metric for tobacco smoking?
Comments on the Quality of English Language
English is acceptable, save for minor issues with convoluted sentences and incorrect syntax. Minor editing is advised.
Author Response
- Lines 20-21: this sentence, which I assume summarizes inclusion criteria for sarcopenia and nutritional issues, is incorrectly written. Please revise to better clarify, adding a reference to the GNRI which I assume is the metric being discussed.
Response:
Thank you for your comment.
We have revised the sentence in lines 20–21 to more clearly summarize the inclusion criteria for sarcopenia and nutritional risk, and we have specified that the Geriatric Nutritional Risk Index (GNRI) was used as the measure for nutritional risk.
Abstract
Aim: This study aimed to investigate how sarcopenia and nutritional risk influence all-cause mortality among older individuals with type 2 diabetes mellitus. Methods: In view of the presence of sarcopenia, defined according to the Asian Working Group for Sarcopenia (AWGS) criteria, and nutritional risk, as determined by the Geriatric Nutri-tional Risk Index (GNRI), a total of 396 participants were divided into four distinct groups (group 1: no nutritional risk and no sarcopenia, n = 306; group 2: nutritional risk and no sarcopenia, n = 32; group 3: no nutritional risk and sarcopenia, n = 36; and group 4: nutritional risk and sarcopenia, n = 22). Mortality risk was assessed through time-to-event analysis using Cox regression.
- Lines 118-121: how were physical activity and smoking habit measured? "Any form of exercise on a weekly basis" is a rather generic catch-all, seeing as it can greatly vary from a mere tranquil walk to endurance training. In a similar fashion, were "packets per year" used as a metric for tobacco smoking?
Response:
Thank you for your valuable comments. Regarding physical activity, we defined it as engaging in any form of exercise at least once per week, regardless of the type, frequency, intensity, or duration of the activity. Due to the retrospective nature of this study, detailed information on the type and amount of exercise was not available. As for smoking history, it was recorded as a binary variable (current smoker or non-smoker). Quantitative data such as pack-years were not collected. We have clarified these points in the revised Methods section.
- Methods
2.2. Data Collection
The earliest of the following was used to determine the duration of diabetes: self-reported diagnosis, the date of the initiation of diabetes-related treatment, or the date of the first abnormal test result related to diabetes. Lifestyle factors—namely physical activity, to-bacco use, and alcohol consumption—were systematically assessed. “Physical activity” was defined as engaging in any form of exercise at least once per week, regardless of the type, frequency, intensity, or duration of the activity. Due to the retrospective nature of the study, detailed information regarding the type and amount of exercise was not available. “Smoking history” was recorded as a binary variable (current smoker or non-smoker). Quantitative data such as pack-years were not collected.